# β-catenin-driven endomesoderm specification is a Bilateria-specific novelty

Tatiana Lebedeva[1,2,7], Johan Boström[3], Stanislav Kremnyov[4], David Mörsdorf [1], Isabell Niedermoser [1,2], Evgeny Genikhovich[5], Andreas Hejnol [4], Igor Adameyko [3,6] & Grigory Genikhovich [1] ✉

Endomesoderm specification by a maternal β-catenin signal and body axis patterning by interpreting a gradient of zygotic Wnt/β-catenin signalling was suggested to predate the split between Bilateria and their sister clade Cnidaria. However, in Cnidaria, the roles of β-catenin signalling in these processes have not been demonstrated directly. Here, by tagging the endogenous β-catenin in the cnidarian *Nematostella vectensis*, we confirm that its oral-aboral axis is indeed patterned by a gradient of β-catenin signalling. Strikingly, we show that, in contrast to bilaterians, *Nematostella* endomesoderm specification is repressed by β-catenin and takes place in the maternal nuclear β-catenin-negative part of the embryo. This completely changes the accepted paradigm and suggests that β-catenin-dependent endomesoderm specification was a bilaterian innovation linking endomesoderm specification to the subsequent posterior-anterior patterning.

Metazoan eggs are polarised along the 'animal-vegetal' axis, with the animal pole being the yolk-poor region where polar bodies are given off in meiosis and towards which the egg nucleus is shifted, and the vegetal hemisphere harbouring most of the yolk. During the early development of bilaterian embryos, β-catenin signalling is involved in two fundamental processes occurring sequentially (Fig. 1a). First, β-catenin accumulates in the cell nuclei at the vegetal side of the embryo to specify the endomesoderm. Then, it forms a nuclear gradient patterning the main, posterior-anterior (P-A) axis[1–11], which is usually roughly colinear to the vegetal-animal axis of the egg. However, the central role of the "canonical" Wnt/ β-catenin (cWnt, Fig. 1b) signalling in the patterning of the main body axis is not restricted to Bilateria. Expression data indicate that cWnt signalling may regulate axial patterning in the earliest branching metazoan groups Ctenophora (comb jellies) and Porifera (sponges)[12–14], and previous functional analyses showed the involvement of the cWnt signalling in the patterning of the main, oral-aboral (O-A) body axis in the bilaterian sister group Cnidaria (sea anemones, corals, jellyfish, *Hydra*)[15–21]. Moreover, nuclear localisation of β-catenin on one side of the sea anemone embryo at the early blastula stage and its failure to gastrulate upon β-catenin loss-of-function suggested that the involvement of β-catenin signalling in the endomesoderm specification was also an ancestral feature conserved at least since before the cnidarian-bilaterian split some 700 Mya[11,15,22,23]. However, despite convincing circumstantial evidence, there was no direct proof for either of these two functions of β-catenin signalling in a cnidarian. Here we set out to obtain such proof by tagging endogenous β-catenin in a model cnidarian—the sea anemone *Nematostella vectensis*—with superfolder GFP[24] and detecting the localisation of nuclear β-catenin at the time of germ layer specification and in the axial patterning phase.

## Results

### Nuclear β-catenin is involved in the O-A patterning

Previously we showed that genes expressed in distinct ectodermal domains along the O-A axis in *Nematostella* gastrula react dose-dependently to different levels of pharmacological upregulation of the

[1]Department of Neurosciences and Developmental Biology, Faculty of Life Sciences, University of Vienna, Vienna, Austria. [2]Vienna Doctoral School of Ecology and Evolution, University of Vienna, Vienna, Austria. [3]Department of Neuroimmunology, Center for Brain Research, Medical University of Vienna, Vienna, Austria. [4]Institute for Zoology and Evolutionary Research, Friedrich Schiller University Jena, Jena, Germany. [5]Engelsa pr. 40-6, St. Petersburg, Russia. [6]Department of Physiology and Pharmacology, Karolinska Institutet, Stockholm, Sweden. [7]Present address: Institute for Zoology and Evolutionary Research, Friedrich Schiller University Jena, Jena, Germany. ✉e-mail: grigory.genikhovich@univie.ac.at

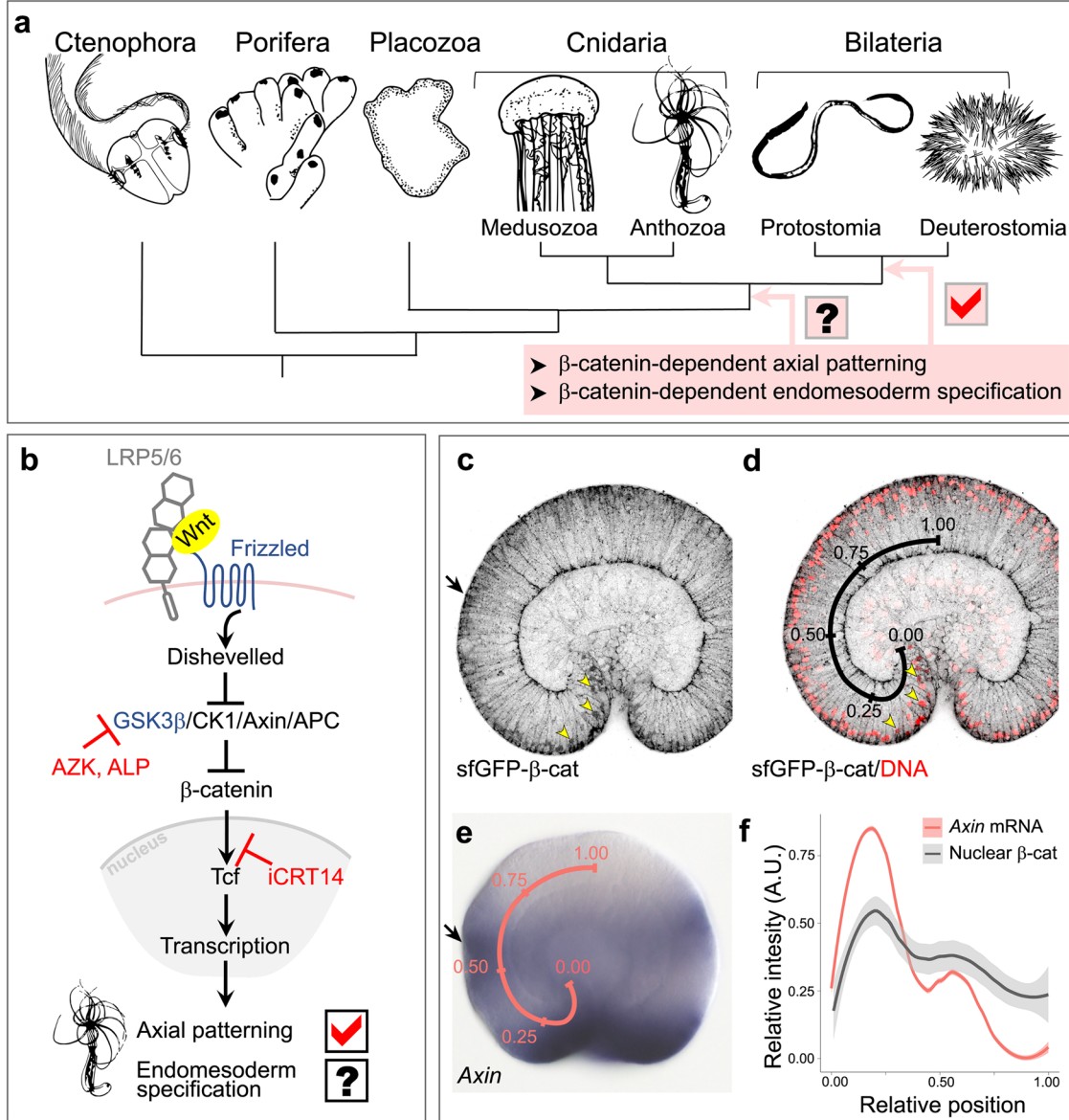

**Fig. 1 | Nuclear sfGFP-β-catenin forms a bimodal oral-to-aboral gradient in late gastrula-stage embryos. a** The questions of this study. **b** Wnt/β-catenin pathway. Upon pathway activation, β-catenin translocates into the nucleus. Azakenpaullone (AZK) and alsterpaullone (ALP) are GSK3β inhibitors; iCRT14 is an inhibitor of the β-catenin/Tcf interaction. **c** Anti-GFP antibody staining detects sfGFP-β-catenin at the cell membranes and in the nuclei. **d** Overlay of the anti-GFP signal with the nuclear staining shows the positions, at which anti-GFP staining was quantified. The first nucleus, where anti-GFP staining intensity was measured is located at the relative position 0.00; the last nucleus – at the relative position 1.00. Yellow arrowheads on

(**c**) and (**d**) point at the same example nuclei. **e** *Axin* in situ hybridisation staining intensity was measured along the pink line from the relative position 0.00 to the relative position 1.00. **f** LOESS smoothed curves show that nuclear sfGFP-β-catenin forms an oral-to-aboral gradient with two peaks (*n* = 6). These peaks correspond to the positions where β-catenin target *Axin* expression peaks as well (*n* = 10). Shaded areas show the 99% confidence interval for the mean. Black arrows on (**c**) and (**e**) point at the second, lower peak of nuclear sfGFP-β-catenin and the corresponding second peak of *Axin* expression.

β-catenin signalling (Fig. 1b[15]). This suggested that β-catenin signalling activity was graded along the O-A axis of the gastrula stage embryo. Downstream, β-catenin signalling activates a set of transcription factors, among which the more orally expressed ones act as transcriptional repressors of the more aborally expressed ones[17]. As a result of these regulatory interactions, the initially ubiquitous aboral identity of the embryo is restricted in a β-catenin-dependent manner to the future aboral domain as the oral and the midbody domains appear and become spatially resolved[17]. In this process, JNK signalling appears to act agonistically with β-catenin signalling: JNK inhibitor treatment aboralizes the embryo, and JNK inhibition is also capable of dose-

dependently rescuing the oralization caused by pharmacological upregulation of β-catenin signalling with the GSK3β inhibitor azakenpaullone (Fig. 1b; Supplementary Fig. 1). The striking similarity of the regulatory logic of the β-catenin-dependent axial patterning and the complement of the downstream transcription factors between the sea anemone and the deuterostome bilaterians suggests the homology of the cnidarian O-A and the bilaterian P-A axis[1,2,6,17,25,26].

To directly verify the presence of the oral-to-aboral β-catenin signalling, we used CRISPR/Cas9-mediated genome editing to generate a knock-in line, in which the nucleotides coding for the first 5 amino acids of β-catenin were replaced by the superfolder GFP (sfGFP)

coding sequence. To test for the presence of the nuclear β-catenin gradient along the oral-aboral axis of the embryo, we incrossed heterozygous F1 polyps (*wild type/sfGFP-β-catenin*) and allowed the offspring to develop until late gastrula stage. As expected, approximately 3/4 of the embryos were fluorescent, however, fluorescent microscopy of live embryos only revealed strong signal at the cell boundaries—in line with the function of β-catenin in the cell contacts[27]. In order to detect nuclear sfGFP-β-catenin, we fixed the embryos and stained them with an anti-GFP antibody. Antibody staining revealed a comparatively weak but clear nuclear signal forming an oral-to-aboral gradient in the ectoderm (Fig. 1c, d). Quantification of the signal intensity in all ectodermal nuclei starting from the deepest cell of the pharyngeal ectoderm (relative position 0.00) and ending with the cell in the centre of the aboral ectoderm (relative position 1.00) showed a peak of nuclear sfGFP-β-catenin in the bend of the blastopore lip (approximately at relative position 0.20), and a second, smaller peak at the border between the midbody and the aboral domain (approximately at relative position 0.60). Both peaks coincide with the peaks of expression of the conserved and highly sensitive β-catenin signalling target *Axin*[15,17,19] (Fig. 1e, f). We then upregulated β-catenin signalling with different concentrations of the GSK3β inhibitor alsterpaullone (Fig. 1b) and showed that changes in the expression domains of the β-catenin-dependent genes *Axin*, *Brachyury* (oral marker), *Wnt2* (midbody marker), and *Six3/6* (aboral marker) are consistent with the changes in the amount of nuclear sfGFP-β-catenin in different positions along the O-A axis (Supplementary Fig. 2). Thus, we conclude that the initial assumption that genes expressed in distinct domains along the O-A axis react to a graded β-catenin signal is indeed correct.

## Nuclear β-catenin does not specify *Nematostella* endomesoderm

Our next goal was to verify the involvement of β-catenin signalling in the specification of the endomesoderm in *Nematostella*. It must be noted that the views on *Nematostella* germ layer identity are currently undergoing a transition. Traditionally, cnidarians, including sea anemones, are described as animals with an outer ectodermal layer, an invaginated ectodermal pharynx (absent in hydroids), and an inner cell layer termed endoderm or endomesoderm in different sources. Although cnidarians demonstrate a striking diversity of gastrulation modes[28], the predominant gastrulation type characteristic for true jellyfish, corals, and sea anemones is invagination of the endomesoderm into a hollow blastula. However, recent studies show very convincingly that cnidarian endomesoderm resembles the bilaterian mesoderm, and cnidarian pharyngeal ectoderm resembles bilaterian endoderm both in terms of gene expression and in cell behaviour[29,30]. A new detailed expression analysis shows that the first zygotically expressed germ layer-specific markers appear at 6 h post-fertilisation in a contiguous patch of cells corresponding to the future endomesoderm[31]. These are mesodermal markers, and the onset of their expression is followed by the first endodermal markers around 8–10 hpf[15,31]. Some of the future key endodermal markers, such *Brachyury*, *Wnt1*, *Wnt3*, and *WntA* are initially transiently co-expressed with the mesodermal markers (Supplementary Fig. 3, see also ref. 15) making it possible to call this transient state a true "endomesoderm". As development continues, the cells of the endomesoderm retain mesodermal marker expression, and eventually undergo apical constriction and invaginate, while endodermal marker expression progressively moves out of the mesoderm and into a ring of cells surrounding the mesoderm—the future blastopore lip/pharyngeal ectoderm (Supplementary Fig. 3). Recent analysis showed that this definitive endodermal domain is induced in the ectodermal cells in a Delta/Notch-dependent manner by ~12 hpf[31]. Based on marker gene expression, henceforth, we will use the terms "endoderm" and "mesoderm" for the pharyngeal ectoderm and endomesoderm, respectively, but we will continue using the term "endomesoderm" for

the territory in the early embryo, which is permissive for the expression of the mesodermal and endodermal markers.

In line with the expression data[31], our earlier pharmacological experiments showed that *Nematostella* endomesoderm specification is an early event happening at or prior to 6 hpf[17,19], at which time nuclear sfGFP-β-catenin starts to be detectable in the developing embryos by fluorescent microscopy. Hence, in order to verify the involvement of β-catenin signalling in the specification of the endomesoderm, we immobilised sfGFP-β-catenin expressing embryos in low melting point agarose and performed live imaging from early cleavage until the onset of gastrulation (Fig. 2a-o, Supplementary Movies 1, 2, Supplementary Fig. 4). Faint nuclear sfGFP-β-catenin signal became detectable as early as 32-64-cell stage and increased in strength from the 128-cell stage on. From the very start, nuclear signal was confined to ~2/3 of the embryo. Nuclear sfGFP-β-catenin was visible during every cell cycle until shortly after the desynchronization of the mitotic divisions at ~2000-cell stage[32], after which it became too weak to be detected by live imaging, while the sfGFP-β-catenin signal in the cell contacts remained strong. Strikingly, in all embryos we live-imaged ($n = 10$), the mesoderm invaginated on the side opposite to where nuclear sfGFP-β-catenin was detectable at earlier stages, i.e. early nuclear sfGFP-β-catenin was always observed on the future aboral side of the embryo. To make sure that we were indeed observing the nuclear sfGFP-β-catenin dynamics, we also live-imaged sfGFP-β-catenin expressing embryos, which were incubated in a 5 μM solution of the GSK3β inhibitor alsterpaullone from fertilisation on (Supplementary Movies 3, 4). In line with the previous publications[11,15], upon GSK3β inhibition, fluorescent signal was observed in all nuclei from 16 to 32 cell stage on, and the embryos failed to gastrulate. In order to independently confirm that strong nuclear sfGFP-β-catenin signal we observed is indeed localised to the ectodermal rather than the endomesodermal domain, we combined hybridisation chain reaction with probes against an early endomesodermal marker *ERG* with anti-GFP antibody staining in 9 hpf and 12 hpf embryos. In 9 hpf embryos (~512-cell stage), strong GFP signal was detected in the interphase nuclei of all the cells except those expressing *ERG*, supporting our live imaging observations (Supplementary Fig. 5). In contrast, at 12 hpf (~1024-2000-cells), the strong aboral sfGFP-β-catenin staining started to progressively disappear and become replaced by weak staining, most prominent on both sides of the *ERG* expression boundary (Supplementary Fig. 5). This nascent nuclear sfGFP-β-catenin signal in the previously β-catenin-negative *ERG*-expressing cells is most likely caused by *Wnt1*, *Wnt3* and *WntA*, which start to be co-expressed in the endomesoderm from 10 hpf on (ref. 15, Supplementary Fig. 3).

Since endomesoderm specification by vegetal nuclear β-catenin has been shown to be controlled by the maternally deposited components in several bilaterians[5,8,33–36], we next tested whether nuclear β-catenin observed in *Nematostella* embryos in the non-endomesodermal domain is maternal as well. To do so, we performed two reciprocal experiments: we crossed a homozygous *sfGFP-β-catenin* female to a wild-type male and a homozygous *sfGFP-β-catenin* male to a wild-type female. Then, we assayed the embryos from both crosses for the presence of the nuclear β-catenin by staining them with anti-GFP antibody at 9 hpf, when strong aboral signal is detectable in live imaging, and at 12 hpf, when nuclear β-catenin stops to be detectable by live imaging but β-catenin-dependent O-A patterning starts. We showed that in the embryos developing from the eggs laid by an *sfGFP-β-catenin* female, strong nuclear anti-GFP signal was observed at 9 hpf (Fig. 2p). In contrast, no nuclear anti-GFP signal was observed at 9 hpf in the embryos laid by the wild type female and fertilised by the *sfGFP-β-catenin* male (Fig. 2q). By 12 hpf, weak nuclear signal was observed in both crosses (Fig. 2r, s). Thus, we conclude that the strong nuclear sfGFP-β-catenin we observed in live imaging is maternal, while weak nuclear signal observed in the 12 hpf blastulae (Fig. 2r, s, Supplementary Fig. 5) as well as gastrulae (Fig. 1c, d) is zygotic.

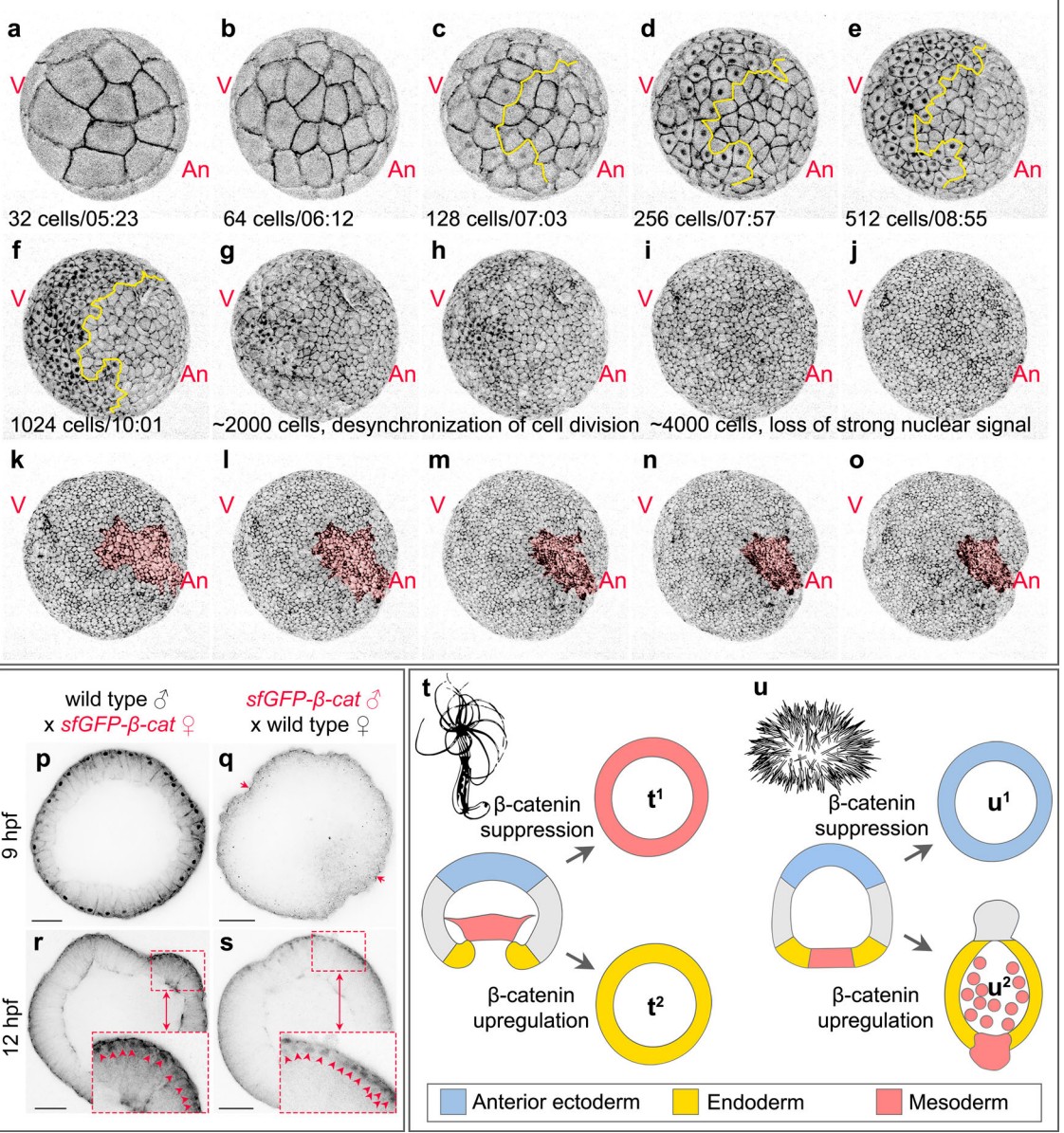

**Fig. 2 | Maternal sfGFP-β-catenin accumulation is observed at the vegetal pole, opposite to the future gastrulation site.** Individual frames from Supplementary Movie 1 showing the same embryo over the course of development. An – animal/oral/posterior pole, V – vegetal/aboral/anterior pole. From the time when mesodermal cells start to show signs of apical constriction, mesoderm is highlighted pink. The embryo is embedded in agarose and does not rotate between (**a**) and (**o**). sfGFP-β-catenin – black signal observed in the cell contacts (**a**–**o**) and in the interphase nuclei in the vegetal/aboral half of the embryo (**c**–**h**). Average developmental times on (**a**–**o**) are taken from ref. 32; time after fertilisation on (**a**–**f**) is shown in hh:mm format. See also Supplementary Fig. 4. Yellow line on (**c**–**f**) demarcates the sharp boundary between the nuclear sfGFP-β-catenin-positive and nuclear sfGFP-β-catenin-negative cells until the loss of synchronicity in cell division on (**g**). **p**–**s** Anti-GFP antibody staining shows that early vegetal nuclear sfGFP-β-catenin is maternal. Arrows on (**q**) show weak sfGFP signal in the cell-cell contacts. Arrowheads on (**r**, **s**) point at sfGFP-β-catenin-positive nuclei. Scale bars 50 μm. The experiment was replicated three times with similar results. **t, u** Comparison of the effect of the downregulation of β-catenin signalling in *Nematostella* and sea urchin. **t¹** Effect of the β-catenin morpholino injection. **t²** Effect of early (<6 hpf) β-catenin signalling activation. **u¹** Effect of *Axin* mRNA injection. **u²** Effect of constitutively active *β-catenin* mRNA injection. The cartoons showing the effects of the up- and downregulation of the β-catenin signalling are based on data from refs. 23,26,33.

## Discussion

Previous analyses of the *β-catenin-GFP* mRNA injected *Nematostella* embryos showed nuclear β-catenin-GFP localisation on one side of the early blastula in the untreated embryos and in all blastoderm cells of the embryos upon GSK3β inhibition[11,15]. Moreover, *Nematostella* embryos with the nuclear localisation of β-catenin directly suppressed by various methods failed to gastrulate, remaining perfect blastula-like spheres[11,23,37]. Morphologically, this effect resembled the gastrulation block caused by functional suppression of β-catenin in sea urchin[5] and

led to the conclusion that the endomesoderm in *Nematostella*, just like the endomesoderm in a number of bilaterians, is specified by an early β-catenin signal at the future gastrulation pole of the embryo[11]. Although universally accepted in the field (also by us—see for example[15,38]), this hypothesis was contradicted by several important observations (Fig. 2t-u², Supplementary Note 1, Supplementary Figs. 1, 5–7). First, unlike sea urchin embryos with suppressed β-catenin, *Nematostella* β-catenin morphants ubiquitously expressed mesodermal markers, rather than the zygotic markers of the aboral/anterior

ectoderm corresponding to the low "β-catenin signalling" end of the O-A axis (compare Fig. 2t[1] with Fig. 2u[1], Supplementary Figs. 6, 7[23,31]; Supplementary Note 1). Second, upon pharmacological activation of β-catenin signalling by early treatment with a GSK3β inhibitor, *Nematostella* embryos failed to segregate the mesoderm, form blastopore lips, and gastrulate. Treated embryos remained spherical and ubiquitously expressed endodermal markers such as *Brachyury*, *FoxA*, *FoxB*, while mesodermally expressed markers *SnailA*, *ERG* and many others were abolished (Fig. 2t[2] [19,23,39–42], Supplementary Figs. 6, 7; Supplementary Note 1). In contrast, in the sea urchin, activation of β-catenin signalling also strongly expands the endodermal portion of the embryo, however, the mesoderm is not lost (Fig. 2u[2]; Supplementary Fig. 6[33]). Taken together, these data allow us to hypothesise that: i) similar to the situation in deuterostome Bilateria[1,2,34,43], activation of β-catenin signalling can induce endodermal marker expression in the ectodermal territory of *Nematostella*, and ii) unlike in the sea urchin and other Bilateria, *Nematostella* endomesoderm specification is repressed by β-catenin signalling rather than activated by it. In line with this hypothesis, expression analysis clearly shows that mesodermal marker genes are repressed by β-catenin signalling (Supplementary Fig. 7[31]). Moreover, the onset of the expression of the endodermal marker genes known to be under positive Wnt/β-catenin regulation only slightly later in development[17,23], is also not activated by β-catenin (Supplementary Fig. 7). The presence of nuclear sfGFP-β-catenin on the aboral rather than on the oral side of the developing embryos (Fig. 2a-o, Supplementary Movies 1, 2) directly supports this hypothesis and explains the discrepancy between the sea urchin and *Nematostella* phenotypes mentioned above (Fig. 2t, u, Supplementary Fig. 6).

Our finding that an oral-to-aboral gradient of nuclear β-catenin exists in the *Nematostella* gastrula confirms a number of previous assumptions on the mode and logic of the oral-aboral patterning in this animal and is in line with the idea that the cnidarian O-A axis corresponds to the bilaterian P-A axis[15,17,19]. However, our second observation that *Nematostella* endomesoderm forms in the β-catenin-negative domain has a much greater importance for the understanding of the early evolution of the body axes and germ layers. In Bilateria, unless physically prevented by large amounts of yolk, endomesoderm specification and gastrulation take place at the vegetal pole, i.e. posteriorly. In contrast, in Cnidaria, gastrulation modes are highly variable. Some species gastrulate by invagination, unipolar ingression or epiboly, while others have multipolar modes of gastrulation such as cellular, morular or mixed delamination or multipolar ingression[28]. In case of multipolar gastrulation, germ layer specification and gastrulation movements are spatially uncoupled from the universally cWnt-dependent O-A patterning[20,44–47]. Importantly, in cnidarians with unipolar modes of gastrulation, endomesoderm specification, and gastrulation always takes place at the animal, rather than at the vegetal pole, and in all cnidarians analysed so far, the animal-vegetal axis of the egg exactly corresponds to the O-A axis of the embryo independent of whether they have a multipolar or a unipolar mode of gastrulation[28,32,48–50]. Previously, it has been proposed that the activation of the β-catenin-dependent endomesoderm specification at the vegetal rather than at the animal pole of a stem bilaterian resulted in the inversion of the position of the gastrulation site in Bilateria[51] (Fig. 3a). Our new data suggest a different scenario (Fig. 3b). Both, in *Nematostella* and in Bilateria, maternal β-catenin accumulates in the vegetal pole nuclei, however, the specification of the endomesoderm by this signal appears to be a Bilateria-specific novelty, which linked germ layer specification, gastrulation movements and P-A patterning (Fig. 1a). In contrast, in Cnidaria, endomesoderm specification appears to be either negatively controlled by β-catenin (as in *Nematostella*) or not to be controlled by β-catenin at all (as in cnidarians with multipolar gastrulation modes), which provides a plausible explanation for the variety of gastrulation modes observed in this phylum. For example, in the unipolarly gastrulating hydroid *Clytia*, endomesoderm formation

is delayed but not abolished by the knockdown of the maternally deposited *Wnt3* or *Frizzled1*, which have been shown to be the central components of the cWnt-dependent O-A patterning in this animal, while O-A patterning is completely disrupted, and the embryo stays spherical[45,46]. Similarly, endomesoderm specification is not affected by the pharmacological suppression of the β-catenin signalling in the multipolarly gastrulating hydroid *Dynamena*[47].

In order to verify that endomesoderm specification by β-catenin is indeed a bilaterian novelty it will be important to address the mechanisms of endomesoderm specification in the representatives of earlier branching animal clades. Among Placozoa, Porifera and Ctenophora (Fig. 1a), only representatives of the latter group have a morphologically distinct gut and clearly gastrulate. Notably, in ctenophores, the oral pole forms at the animal pole, like in Cnidaria[52]. Due to difficulties with microinjection, gene function studies in the model ctenophore *Mnemiopsis leidyi* are still in their infancy[53,54]. Nevertheless, the role of β-catenin has been addressed in a preprint by Salinas-Saavedra and colleagues where they showed that injection of the mRNA of the cytoplasmic domain of the sea urchin cadherin as well as CRISPR/Cas9-mediated β-catenin knockout resulted in the loss of gut and musculature[55]. However, our pharmacological tests show a different picture. Previous analyses demonstrated that LysoTracker is a robust marker of the forming gastrovascular system in *Mnemiopsis*[53]. Treatment of the developing *Mnemiopsis* embryos from 8 to 16 cell stage on with the inhibitor of the β-catenin/Tcf interaction iCRT14 (Fig. 1b) resulted in the formation of the cydippid stage juveniles with shortened pharynx and expanded gut (by analogy to *Nematostella*–a putative endodermal and mesodermal tissue, respectively; Fig. 3c), suggesting that, similar to the situation in cnidarians, β-catenin may play an "antiendomesodermal" role in this ctenophore.

In summary, our work resolves previous conflicting findings regarding the early functions of β-catenin signalling in the sea anemone *Nematostella*: we show that maternal β-catenin signalling inhibits (rather than activates) endomesoderm specification, while a gradient of zygotic β-catenin signalling patterns the endoderm and ectoderm along the O-A axis. We propose that the axial patterning function of Wnt/β-catenin is conserved between Cnidaria and Bilateria, however, endomesoderm specification by β-catenin appears to have evolved in Bilateria and possibly tethered gastrulation to the vegetal pole of the embryo making it posterior (Fig. 1a). Many questions arise based on our observations: i) what causes endomesoderm specification and subsequent gastrulation movements in the β-catenin-negative domain in *Nematostella* and in random positions in cnidarians with multipolar gastrulation modes, ii) what regulatory changes tethered bilaterian gastrulation to the ancestral site of the nuclear β-catenin accumulation at the vegetal pole, or iii) does β-catenin signalling really prevent endomesoderm formation in the earlier branching Ctenophora, which also gastrulate from the animal pole? In any case, our data suggest that the current view on the ancestral mode of endomesoderm specification in animals needs to be re-assessed.

## Methods

### Animal culture, generation of the sfGFP-β-catenin knock-in line

*Nematostella vectensis* polyps were maintained and spawning was induced as described[56]. Embryos were raised at 21 °C. To generate a *sfGFP-β-catenin* knock-in, a single gRNA with a protospacer 5′-ACCATGGAGACACACGGTAT-3′ recognising a sequence starting at the position 134602 on the scaffold_183 of the *Nematostella* genome v.1[57] was selected using CHOPCHOP[58], and CRISPR/Cas9 genome editing was performed as described in ref. 59. For homologous recombination, we generated a fragment in which the first five triplets of the *Nematostella β-catenin* coding sequence were replaced with the *Superfolder GFP* coding sequence introduced in frame by Gibson assembly. The fragment containing the homology arms and *sfGFP* was amplified using the primers 5′-GTGGAATTCGCAGCATTTCTCA-3′ and 5′-TCAAGG

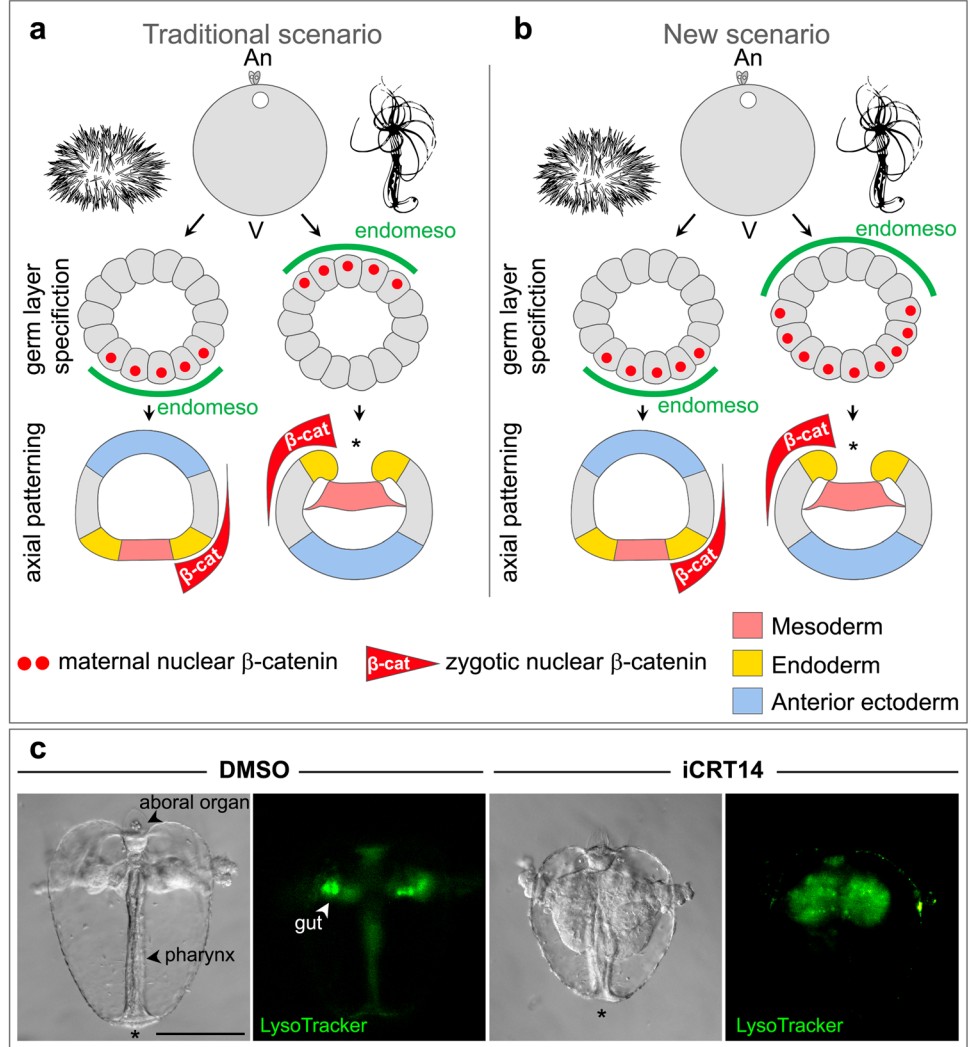

**Fig. 3 | β-catenin signalling may have been co-opted for endomesoderm specification at the base of Bilateria.** Traditional scenario of the β-catenin-dependent germ layer specification and main body axis patterning (**a**) and its modification based on our findings (**b**). An – animal pole; V – vegetal pole. **c** iCRT14 treatment shortens the pharynx and expands the gut (green staining) in the ctenophore *Mnemiopsis*. The experiment was replicated three times with similar results. Scale bar 50 μm. Asterisks denote the positions of the blastopore.

ATGGCTCAGCAAGC-3′, which were modified as described[60]. F0 animals with clear fluorescent patches were raised to sexual maturity and crossed to wild type to generate heterozygous F1. To confirm the knock-in, we clipped single tentacles from individual heterozygous F1 animals, extracted genomic DNA from them and performed PCR using the primers 5′-GGTCGTAGATGGTACCCTAAG-3′ and 5′-CAACTCTGG-GATAGCACGTGTAG-3′ located in the *β-catenin* genomic locus upstream and downstream of the homology arms. This PCR resulted in two *β-catenin* locus fragments with and without the *sfGFP* insertion, which we confirmed by Sanger sequencing. Genotyped knock-in animals were raised to maturity, sexed, and intercrossed. The offspring of these genotyped F1 animals was used in the experiments. All inhibitor treatments were performed as described in ref. 17. For morpholino-mediated knockdown of *Nematostella* β-catenin, we microinjected a previously described[23] translation-blocking morpholino TTCTTCGACT TTAAATCCAACTTCA. To knock down *LRP5/6* by RNAi, we electroporated a previously described[19] shRNA against the target sequence GAGAGCCTTCCACTTGTAA according to the published protocol[37].

Sea urchin experiments were performed during the University of Vienna course 'Developmental Biology of Marine Invertebrates in Villefranche-sur-Mer'. Mature male and female *Paracentrotus lividus* were kindly provided by the Croce and Schubert lab and handled as described by ref. 61. For documenting early development (Supplementary Fig. 6c, d), fertilisation envelopes were removed right after fertilisation. For later stages (Supplementary Fig. 6e, f), the embryos were allowed to develop inside the fertilisation envelopes. DMSO and 1 μM azakenpaullone treatments we started after fertilisation and continued until fixation.

Laboratory cultures of cydippid *Mnemiopsis leidyi* were kept in 3 L Kreisel tanks at 17–19 °C on a 17 h/7 h light/dark cycle in 27.5 ppt artificial seawater (ASW) and spawned ~6–7 h post-darkness. Animals used to set up culture were originally collected at the Kristineberg Marine Research Station, Sweden. For embryo collection, sexually mature cydippids were transferred into 200–250 ml beakers in the evening and screened for mature eggs and early embryos next morning. To inhibit β-catenin signalling, 8–16 cell embryos were treated with 10 μM iCRT14 (SML0203, Sigma). At ~30 hpf, the embryos were transferred into filtered ASW containing 1 μM LysoTracker Yellow HCK-123 (Molecular Probes) and 10 ng/μl Hoechst 33342 for at least 1 h at room temperature. Live embryos were then mounted on glass slides in ASW for imaging with a Zeiss Axioplan microscope equipped with a Zeiss AxioCam camera.

## Gene expression analysis, antibody staining and staining intensity measurements

For qPCR, total RNA was extracted with Trizol (Invitrogen) from 9 hpf and 12 hpf treated and control embryos in biological triplicates and reverse transcribed using LunaScript RT SuperMix Kit (NEB). qPCR results were normalised to *GAPDH*. Primer sequences can be found in Supplementary Table 1.

For anti-GFP antibody staining, the embryos were fixed for 1 h in 4%PFA/PBS-TT (PBS-TT = 1x PBS containing 0.2% Tween20 and 0.2% TritonX100) at 4 °C, washed five times for 5 min in PBS-TT, incubated for 2 h in a blocking solution consisting of 95% BSA/PBS-TT and 5% heat inactivated sheep serum (BSA/PBS-TT = 1% BSA w/v in PBS-TT), and stained overnight at 4 °C in rabbit polyclonal anti-GFP (abcam290, RRID:AB_303395) diluted 1:500 in the blocking solution. Unbound antibody was removed by five 15 min washes in PBS-TT, then the embryos were blocked again and stained overnight at 4 °C with AlexaFluor488 donkey anti-rabbit IgG (ThermoFisher A32790, RRID:AB_2762833) diluted 1:1000 in the blocking solution. The unbound secondary antibody was removed by five washes with PBS-TT; DAPI was added to the first wash to counterstain the nuclei, then the embryos were gradually embedded in Vectashield (Vectorlabs). 16-bit images of the DAPI and anti-GFP staining were obtained using the Leica SP8 LSCM equipped with a 63x glycerol immersion objective ($n = 6$). Single and double in situ hybridisation with RNA probes against *Nematostella Axin, Brachyury, Wnt1, Wnt2, Wnt3, WntA, ERG*, and *Six3/6* was performed as described previously[17].

The anti-GFP staining intensity was measured over all ectodermal DAPI-positive nuclei starting from the deepest pharyngeal cell (relative position 0.00−see Fig. 1D) to the cell in the middle of the aboral ectodermal domain (relative position 1.00−see Fig. 1d) using FIJI[62]. Briefly, to identify the ROIs, polygonal selections were drawn to separate the pharynx ectoderm and the outer ectoderm based on DAPI signal. Masks were then generated separately for the outer ectoderm and the pharynx ectoderm parts of the image using the *Convert to mask* and the *Watershed* commands. To generate the ROIs for the nuclei, particle analysis with a minimum size of 1 µm² was performed. The resulting ROIs were then manually checked and sorted such that they were arranged from the relative position 0.00 to the relative position 1.00. The mean intensities in the sfGFP channel were measured for all ROIs. The relative position of each nucleus was determined as a nucleus number divided by the total number of nuclei with measured anti-GFP staining intensity in this particular embryo (see Fig. 1D). To normalise the staining intensity across embryos, the relative staining intensity Y in the nucleus k was determined as:

$$Y_k = \frac{y_k - y_{min}}{y_{max} - y_{min}} \qquad (1)$$

Where $y_k$ is the measured nuclear anti-GFP staining intensity over the nucleus $k$, $y_{min}$ is the minimal nuclear anti-GFP staining intensity measured in this particular embryo, and $y_{max}$ is the maximal nuclear anti-GFP staining intensity measured in this particular embryo. The relative anti-GFP staining intensity measurements for all embryos can be found in the Supplementary Data File 1 in the Source Data. *Axin* in situ staining intensity was measured in FIJI[62] on in situ images ($n = 10$) along a smoothened segmented line drawn from the deepest pharyngeal cell (relative position 0.00) to the cell in the middle of the aboral ectodermal domain (relative position 1.00). The relative position of each staining intensity measurement point was determined as a distance from the position 0.00 along the abovementioned smoothened segmented line divided by the total length of this line in the particular embryo (see Fig. 1e). The relative *Axin* staining intensity Y in the position k was determined using the formula [1] above. In this case, $y_k$ is the measured *Axin* in situ staining intensity at the position $k$, $y_{min}$ is the minimal *Axin* in situ staining intensity measured in this

particular embryo, and $y_{max}$ is the maximal *Axin* in situ staining intensity measured in this particular embryo. The relative *Axin* in situ hybridisation staining intensity measurements for all embryos can be found in Supplementary Data File 2 in the Source Data. Since in Supplementary Fig. 2 we are comparing staining in different experimental conditions, no staining intensity normalisation has been performed, and raw data have been plotted. The measurements for all the conditions plotted in Supplementary Fig. 2 can be found in Supplementary Data File 3 in the Source Data.

An aliquot of the Endo1 antibody recognising sea urchin embryonic endoderm[63] was a kind gift of the Croce/Schubert lab at Villefranche. For Endo1 staining, sea urchin embryos were fixed in 3.7% formaldehyde in sea water for 1 h at 4 °C, washed 5 times for 5 min in PTw (1x PBS, 0.1% Tween 20), blocked in 1%BSA/PTw for 2 h and stained overnight in Endo1 antibody (1:15) dissolved in 1%BSA/PTw. Next day, the embryos were washed 5 times 10 min with PTw, blocked as above and stained for 2 h at room temperature with goat anti-mouse AlexaFluor488 secondary antibody (ThermoFisher, A32723, RRID:AB_2633275) at 1:1000 dilution. AlexaFluor568-phalloidin and DAPI were added to the secondary antibody. After five washes, the embryos were embedded in glycerol and imaged using Leica SP8 LSCM.

## HCR/antibody staining combination

HCR was performed according to ref. 64, with modifications. In brief, the HCR probe pool for the fluorescent in situ mRNA visualisation of *Nematostella ERG* was generated using the modified HCR 3.0 in situ probe generator[65] (Supplementary Table 2). The sfGFP-β-catenin knock-in embryos at 9 hpf and 12 hpf were fixed with 4% PFA diluted in 1xPBS for 1 h at room temperature, washed three times for 5 min with PTw (1x PBS, 0.1% Tween 20) and stored in 100% methanol at −20 °C until further use. For rehydration, embryos were washed for 5 min in 70% methanol/30% PTw, then 30% methanol/70% PTw, followed by three washes in PTw. Rehydrated embryos were washed three times for 5 min in 5xSSCT (5x saline-sodium citrate buffer, 0.1% Tween-20). Prehybridization was performed by incubating embryos in Probe Hybridisation Buffer (Molecular Instruments) for 30 min at 37 °C. For hybridisation, embryos were transferred into hybridisation buffer containing the in situ probe set (0.8 pmol in 100 µL hybridisation buffer) and incubated overnight at 37 °C. Samples were then washed three times for 10 min and twice for 30 min in Probe Wash Buffer (Molecular Instruments) at 37 °C. After five 5-min washes in 5xSSCT at room temperature, embryos were transferred to Signal Amplification Buffer (Molecular Instruments) for 30 min. Hairpins h1 and h2 (12 pmol) were heated separately to 95 °C for 90 s, snap-cooled to room temperature in the dark, and added to 100 µL Probe Amplification Buffer. Embryos were then transferred into amplification buffer containing hairpin amplifiers and incubated overnight at room temperature. After amplification, samples were washed in 5xSSCT: three times for 5 min and twice for 30 min. Then, embryos were stained with anti-GFP antibody to detect sfGFP-β-catenin as described above, counterstained with 5 µg/ml DAPI, and gradually embedded in Vectashield (Vectorlabs) for subsequent imaging using Leica Stellaris 5 CLSM.

## Live imaging

Embryos were embedded in 0.7% low-melting agarose in *Nematostella* medium (*Nematostella* medium = 16‰ artificial seawater, Red Sea Salt) in an optical bottom 35 mm Petri dish (D35-20-1.5-N, Cellvis, US) and imaged with a 20X CFI Plan Apo Lambda Objective (Nikon, Japan) using a Nikon Ti2-E/Yokogawa CSU W1 Spinning Disk Confocal Microscope. A 488 nm laser was used in conjunction with a 525/30 Emission Filter (BrightLine HC, Semrock, US) and a 25 µm pinhole size disk. Images were acquired every 5 min using automated imaging, over 25 Z-sections covering 120 µm depth. In the first

experiment, the embryos were left to develop in 16‰ artificial seawater. In the second experiment, the embryos were developed in a 5 μM solution of the GSK3β inhibitor alsterpaullone (Sigma) from 1 hpf on. Live imaging was stopped after gastrulation was observed in the sample developing in the absence of alsterpaullone. Since embryos placed into a GSK3β inhibitor do not gastrulate, we continued the imaging of the alsterpaullone-treated embryos for an additional hour in comparison to the normal embryos. During imaging, the medium in the sample dish was continuously pumped through a tube submerged in a room temperature (-23 °C) water bath to offset heating from the microscope. This was achieved with a modified sample dish lid with two liquid connectors, and a peristaltic pump (Minipuls 3, Gilson, US). Ten embryos were imaged together in each experiment. After imaging, each Z-stack of images was converted into a maximum-intensity projection. The frames presented in Fig. 2a-o with average developmental times for each developmental stage are additionally presented in Supplementary Fig. 1 with actual time stamps from the confocal microscope.

### Reporting summary
Further information on research design is available in Nature Portfolio Reporting Summary linked to this article.

## Data availability
All data are available in the main text or the supplementary materials. Source data are provided with this paper.

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

## Acknowledgements

Sea urchins and Endo1 antibody were kindly provided by the Croce/Schubert lab at the Institut de la Mer de Villefranche. We thank Max Schwab for help with the double in situ in *Nematostella*, the students of the practical course Developmental Biology of Marine Invertebrates in Villefranche-sur-Mer for performing sea urchin inhibitor treatment experiments and the Core Facility for Cell Imaging and Ultrastructure Research of the University of Vienna for the access to the confocal microscopes. The work in Genikhovich group is funded by the Austrian Science Fund (FWF) grants DOI 10.55776/P30404, 10.55776/P32705, and 10.55776/P36080 (to GG). For the purpose of Open Access, the author has applied a CC BY public copyright license to any Author Accepted Manuscript (AAM) version arising from this submission. The work in the Adameyko group is funded by the ERC Synergy grant "KILL-OR-DIFFERENTIATE" 856529. T.L. was a recipient of the Ph.D. completion grant of the Vienna Doctoral School of Ecology and Evolution. D.M. was a recipient of the Lise-Meitner Fellowship M3291-B of the FWF. The work in the Hejnol group is funded by the Human Frontier Science Programme (HFSP) grant RGP0041/2022 (to A.H.).

## Author contributions

T.L. planned and performed experiments and generated the knock-in line; J.B. performed live imaging; S.K. performed iCRT14 treatment in *Mnemiopsis*; I.N. performed *Axin* in situ hybridisation; D.M. and E.G. measured and analysed the gradient data; A.H. provided funding for the *Mnemiopsis* part of the project; I.A. provided access to the spinning disk confocal microscope; G.G. conceived the project, planned and performed experiments, and wrote the paper. All authors edited the paper.

## Competing interests

The authors declare no competing interests.
