## [Transparent Peer Review file · Nature Communications]

β -catenin-driven endomesoderm specification is a Bilateria-specific novelty

Corresponding Author: Dr Grigory Genikhovich

Version 0:

Reviewer comments:

Reviewer #1

(Remarks to the Author)

Manuscript background information

In their manuscript " β -catenin-driven endomesoderm specification is a Bilateria-specific novelty," Lebedeva et al. reexamined roles of Wnt/ β -catenin signaling in body axis formation of the sea anemone *Nematostella vectensis*. Classically, we have thought that both the bilaterians and cnidarians form endomesoderm in response to a Wnt/ β -catenin signal. This signal occurs at the vegetal pole in the bilaterians and at the animal pole in cnidarians such as *Nematostella*. Lebedeva et al. provide exciting new evidence that this interpretation is flawed. Instead, they show that maternal Wnt/ β -catenin signaling occurs at the vegetal pole in both groups, but this signal does not specify endomesoderm in *Nematostella*. Instead, only cells that lack the maternal Wnt signal form endomesoderm. Although the mechanism responsible for endomesoderm specification differs in cnidarians, later axial patterning of the oral-aboral axis is regulated by zygotic Wnt/ β -catenin gradient, just as it is in bilaterians. This paper provides critical information that will significantly modify views of how early development evolved in the animal kingdom and suggests that endomesoderm specification by Wnt/ β -catenin signaling may be an evolutionary innovation limited to bilaterians. This manuscript will be highly interesting to those investigating how cell signals influence evolution of animal body plans.

Comments for transmission to the authors

I appreciated having the opportunity to review this paper. I found it exciting and think that it makes an important novel contribution to evodevo studies. The authors have been thorough in their investigations and show an impressive knowledge of the literature related to this study. As I mentioned above, this paper provides two especially noteworthy contributions. First, the authors have demonstrated that endomesoderm forms at the pole opposite of the cells in which maternal nuclear β -catenin signaling is occurring, therefore the maternal Wnt signal is not responsible for specifying endomesoderm in this group. Second, they demonstrate that a gradient of zygotic β -catenin signaling is influencing cell fates in the specification of the aboral-oral axis in a manner similar to bilaterians. Both findings involved careful analysis and this group seems to have done this well using sound methodology and meeting the expected standards in my field. In most cases, I thought that the authors did a nice job supporting their observations, that their review of the literature was thorough, and that the paper was clearly written. I've made a few suggested improvements below. I sometimes found this study confusing because at least 12 markers and numerous perturbation methods were used. Following this was sometimes difficult so I've included suggestions to help readers with this challenge.

- Line 25, page 2...you said that JNK signaling appears to act agonistically with beta-catenin signaling. Is that what you meant or did you mean that the JNK inhibitor acts agonistically with beta-catenin signaling, since the JNK inhibitor counteracts the effects of activated Wnt signaling?
- When you first mention use of azakenpaullone (line 28, page 2), alsterpaullone (line 18, page 4) and iCRT14 (line 28, page 9) it may help to also refer the reader to Figure 1b. Or at least do this when you mention the first inhibitor, azakenpaullone.
- Lines 11-20, page 7...Elaborate on the specific evidence that contradicted those findings by mentioning the specific markers that were expressed for each group discussed (e.g. Snail, ERP, etc.). In Line 13, why not say ectodermal markers instead of "zygotic markers of the aboral/anterior/low β -catenin-signalling end of the O-A axis?" In the sea urchin the markers expressed would be ectodermal.
- Line 29, page 9...Elaborate a little on why the presence of a shortened pharynx and expanded gut in cydippids suggests an "antiendomesodermal" role for beta-catenin to make it easier for the reader to understand why you came to this

conclusion.

- When discussing the distribution of mesodermal gene expression, you mentioned *ERG* and *snail*. It might be good to also cite two of the *Snail* papers for readers interested in a closer look at *snail* distribution. I'd cite Fritzenwanker, Saina, and Technau, 2004. Analysis of forkhead and *snail* expression reveals epithelial–mesenchymal transitions during embryonic and larval development of *Nematostella vectensis*, as well as Magie, Daly, and Martindale, 2007. Gastrulation in the cnidarian *Nematostella vectensis* occurs via invagination not ingression. Both provided additional helpful information about expression of this mesodermal marker.
- Figure 2... In 2a, remind the reader that they're looking at the animal (oral) and vegetal (aboral) poles (under "V" you could write (aboral) and under "An" you could write (oral)). Because it's necessary to switch back and forth from saying aboral/oral, animal/vegetal, and posterior/anterior, it can be confusing. This modification may help the reader a little.
- Figure 2... What stages are shown in Figure 2k-o? In 2a-j, you clearly indicate the age of the embryo and what time the photo was taken. Please also do this for the figures below those. Also, please explain notable features in f-j and k-o (I believe k-o shows cells that are forming mesoderm. How did you decide which cells were mesodermal in k-o?).
- Extended Data Figure 1... Remind the reader that JNK-IN-8 is a JNK inhibitor and explain how you to expect it to affect Wnt signaling in the caption. Also, for all of the figures include the age or stage of the embryos shown and mention the temperature that embryos were cultured at in the Methods.
- In the figure legends, please clearly define what the markers are as you did in Extended Data Figure 3. There are so many markers used in your paper and it can be difficult to remember which tissues each one marks. I spent a lot of time looking this information up for the different figures. For example define the markers in Extended Data Figure 1... In the paper, you mentioned that *Bra* is an oral marker, *Wnt2* is a midbody marker, and *Six3/6* is an aboral marker. Also tell the readers what *snailA* and *foxA* indicate.
- In Extended Data, Fig. 4c-f, do all 4 photos show sea urchin embryos? What species is it? Readers unfamiliar with these embryos may get confused and think they're comparing *Nematostella* to sea urchin.
- Extended Data, Figure 5... State that the embryo in the figure is *Nematostella* (I assume it is, but this wasn't mentioned). Also, in the Wikramanayake 2003 paper, the authors blocked Wnt signaling by overexpressing cadherin. It would have been nice if you had used the same perturbation in this paper or explained why you chose to use RNAi *LRP5/6*. I expect both perturbations will produce the same phenotype but would have found it more convincing if the same perturbation had been used. Include citations supporting this as a method of blocking Wnt signaling and address whether this perturbation is expected to produce the same phenotype that overexpressing cadherin would have.

Reviewer #2

(Remarks to the Author)

The manuscript 'β-catenin-driven endomesoderm specification is a Bilateria-specific novelty' by Lebedeva et al reinvestigates and reassesses the role of beta-catenin in early endoderm specification of the sea anemone. Intriguingly, they observe an early aboral nuclear beta-catenin localization opposite to the oral 'endomesoderm'. This finding may suggest that β-catenin represses endomesoderm specification in the sea anemone, rather than activating it as seen in bilaterians challenging the established paradigm of beta-catenin function in the cnidarian/bilaterian ancestor.

The authors used CRISPR-Cas9-mediated genome editing to generate a knock-in line tagging the endogenous beta-catenin gene with a Nterminal superfolder GFP in the sea anemone. They used these transgenic animals to first show that there is indeed a weak oral to aboral bimodal beta-catenin gradient at the gastrula stage that correlates with axin expression, responds to different concentration of the GSK3 inhibitor (ALP), and leads to expression changes of target genes. These findings are consistent with previous work and the assumption that an oral to aboral beta-catenin signalling gradient controls domains along the O-A axis.

The authors then focused on the earlier event during the time of endomesoderm specification during the mid-blastula stage. They showed by in situ in mid-blastula stages a domain of cells that coexpress mesodermal (*erg*) and endodermal (*bra*, *wnt1*, *wnt3*, *wntA*) markers. Surprisingly, in video recordings of immobilized embryos at earlier stages nuclear sfGFP-beta-catenin was visible in a large domain opposite to the later oral side of gastrulation and mesoderm formation. Early incubation with the GSK3beta inhibitor (ALP) led to strong nuclear localization of sfGFP-beta-catenin in all mid blastula cells.

Furthermore, crossing reciprocally homozygous sfGFP-beta-catenin female or male to a wildtype male or female, respectively, the authors could demonstrate that the early aboral nuclear expression is indeed maternal, and the later O to A gradient zygotic.

Based on these patterns the authors conclude that early beta-catenin signaling happens aborally/vegetally apparently opposite to the area of endomesoderm specification on the oral side. This pattern is obviously not consistent with the previous hypothesis that endomesoderm specification is specified by an early beta-catenin patterning as observed in many bilaterian embryos.

The authors then argue that previous experimental observations and the observed early aboral pattern of nuclear beta-catenin support rather a mode of early beta-catenin repression of endomesoderm and later mesoderm specification in the sea anemone.

Although the authors show beautifully the nuclear patterning of beta-catenin by using a state of the art knock in tagging strategy, and show convincingly an early aboral pattern of nuclear beta-catenin localization contradicting the currently assumed role of beta-catenin in endomesoderm specification in cnidarians, I have several concerns in regard to some of the interpretation, and the scope and strength of the experimental support. If one wants to make a paradigm changing statement, the authors need to show stronger evidence for endomesoderm repression by early beta-catenin.

Main concern 1:

Indeed I do agree that the beta-catenin morphant phenotype described in Figure 2 t1 (circumferential mesoderm marker expression) might consistent with this new repression model but I don't understand how it can be consistent with the early beta-catenin activation phenotype in Figure 2 t2 (ubiquitous endoderm marker expression)? Would one not expect no mesoderm, and no endoderm upon early beta-catenin activation? If the scenario presented in Fig 3b for the sea anemone is right, should not early beta-catenin activation lead to a complete loss of endomesoderm, and an expansion of the anterior ectoderm/aboral marker genes similar to the beta-catenin suppression phenotype in sea urchin (Figure 2 u1)? Similarly in the discussion, quote: "Finally, the fact that β -catenin inhibits endomesoderm and, subsequently, mesoderm specification rather than activates it explains why upon treatment with a GSK3 β inhibitor prior to 6 hpf, the whole embryo acquires endodermal rather than mesodermal or mixed endodermal and mesodermal fate." I do not understand this conclusion. Should not - based on regulatory logic of a repression model – as a consequence of beta-catenin activation by early GSK inhibition, the embryo be mostly expanding the aboral area, and not expanding the endoderm? How can a ubiquitous repression of endomesoderm specification lead to endoderm? See also extended figure 4: a4 makes sense, but not the a3 result?

Main concern 2:

In this context I also find the description in the abstract too strong "we show that, in contrast to bilaterians, *Nematostella* endomesoderm specification is repressed by β -catenin". While I agree with the reinterpretation of the beta-catenin morphant phenotype, although there is the possibility that beta-catenin morpholinos did not interfere with maternal beta-catenin protein, the authors need to show much stronger experimental evidence for repression with additional markers in early mid blastula stages. Are there any early aboral markers that could be used, and is there a way to elicit and/or inhibit beta-catenin activation in distinct early blastomeres, and look at the response of marker genes? There is no direct evidence for repression.

Main concern 3:

To support the authors hypothesis that endomesoderm specification by beta-catenin is a bilaterian novelty the authors show one ctenophore experiment. The authors argue that an early pharmacological inhibition of beta-catenin in a ctenophore leads to more gut/endodermal tissue consistent with their repression hypothesis. Indeed, the internal tissues including the Lyotracker positive endoderm seem to be expanded after treatment. However, this is by all means fairly weak supporting evidence, and an isolated experiment, that need to be corroborated by more evidence e.g. additional markers, and experiments. I don't think it should be used at this stage as supporting evidence. This phenotype needs to be characterized better.

Minor concerns:

1. Figure 2 a to o: How do the authors discern the animal and vegetal pole here? Based on the location of polar bodies, or retroactively based on the later animal location of the mesoderm? Is the pink area in later stages (prospective mesoderm) discerned based on morphology and shape of the cells? What is the localization of the polar bodies (also in the movies)? The authors should show early nuclear beta-catenin localization together with early endomesodermal and/or anterior ectoderm markers.

2. Figure 2 p: why is there circumferential localization of beta-catenin? It should be absent on one side. I assume the authors will argue that it is the section but I would prefer a section that shows both the activation only in one part but not the other as less misleading.

3. Figure 2c: The weak nuclear beta-catenin is difficult to discern from this section e.g. one can see some nuclei on left side of the gastrula but not on the right. Why didn't the authors normalize the nuclear GFP signal using nuclear DNA staining?

3. Extended Figure 3: d to g

The ERG positive territory in the 12hpf sea anemone embryo has a very different appearance and shape in each stained embryo. Would one not expect a more regular shaped appearance of this domain based on similar endomesoderm-like territories in other embryos? It is irritating that the orientation for each double in situ is all over the place. This irregular patch of 'endomesoderm' in wildtype sea anemone seems kind of strange. However, I am also not that familiar with *Nematostella* embryos.

4. Extended Figure 3: d to g, c to h

It is somewhat unclear how the early 'endomesoderm' patch relates to later mesoderm and endoderm in *Nematostella*. Does the endomesoderm area differentiate into mesoderm in gastrula stages or into both the endodermal 'ring' and the mesoderm? Or is the endoderm marker expression somewhat outside the earlier 'endomesoderm' patch?

5. Extended data Fig. 2: It looks like the size of embryos was really affected by the treatment (compare h to i).

6. Lane 24: 'Midbody' domain: misleading term consider other name

7. Page 2 [15-20] Rephrase this complex sentence.

8. Page 4 [15] Point out the small peak at the border between the midbody and the aboral domain (approximately at relative position 0.60) in Fig 1e.

Reviewer #3

(Remarks to the Author)

In this article, Lebedeva et al. analyzed the role of b-catenin signaling in endomesoderm specification and axial patterning in the Cnidaria *Nematostella vectensis*. They characterized b-catenin localization by tagging the endogenous protein with GFP and also tested the consequences of up- or downregulation of b-catenin signaling. Their data confirm the role of b-catenin in axial patterning that was previously suggested. More interestingly, they observed that b-catenin is excluded from the early endomesoderm territory, contrary to what was previously assumed, suggesting that b-catenin driven endomesoderm specification is a novelty of Bilateria.

The article is overall well written and the data are convincing. However, a few points could be improved before publication:

- the authors claim that *Nematostella* endomesoderm is repressed by b-catenin. However, I think the authors could provide better evidence to support this claim. The fact that b-catenin is excluded from the early endomesoderm territory doesn't necessarily mean that b-catenin is repressing it. They show the effect of b-catenin depletion or upregulation on the expression of markers at a late stage when mesoderm and endoderm are already separated. b-catenin depletion (b-catenin MO) expands mesoderm but it also represses endoderm. b-catenin upregulation (AZK) represses mesoderm but it also expands endoderm. To me these observations at a late stage make the interpretation in term of "endomesoderm specification" a bit difficult. The authors nicely define endomesoderm at 12 hpf blastula stage (an earlier stage) as a territory that coexpresses endodermal and mesodermal markers (extended data Fig. 3d-g). I think they should analyze the effect of b-catenin depletion or upregulation on the co-expression of these markers at this stage (12 hpf blastula).

- it would be useful to have a definition of the terms "animal-vegetal axis" and "oral-aboral axis" at the beginning of the manuscript.

Version 1:

Reviewer comments:

Reviewer #1

(Remarks to the Author)

I approve of the changes that were made and appreciated the authors' explanations concerning why they chose not to modify in a few cases (I support their decisions). I look forward to seeing this manuscript published in the journal.

Reviewer #2

(Remarks to the Author)

I thank the authors for carefully and patiently addressing my previous concerns, and putting such a clear, rigorous, and exciting revision together. No objections and concerns.

Minor error:

Page 6 line 17: much greater impact?

However, our second observation that *Nematostella* endomesoderm forms in the β -catenin negative domain has a much greater import for the understanding of the early evolution of the body axes and germ layers.

Reviewer #3

(Remarks to the Author)

The authors have addressed all the points that I raised in a satisfactory manner.

REVIEWER COMMENTS

Reviewer #1 (Remarks to the Author):

Manuscript background information

In their manuscript “ β -catenin-driven endomesoderm specification is a Bilateria-specific novelty,” Lebedeva et al. reexamined roles of Wnt/ β -catenin signaling in body axis formation of the sea anemone *Nematostella vectensis*. Classically, we have thought that both the bilaterians and cnidarians form endomesoderm in response to a Wnt/ β -catenin signal. This signal occurs at the vegetal pole in the bilaterians and at the animal pole in cnidarians such as *Nematostella*. Lebedeva et al. provide exciting new evidence that this interpretation is flawed. Instead, they show that maternal Wnt/ β -catenin signaling occurs at the vegetal pole in both groups, but this signal does not specify endomesoderm in *Nematostella*. Instead, only cells that lack the maternal Wnt signal form endomesoderm. Although the mechanism responsible for endomesoderm specification differs in cnidarians, later axial patterning of the oral-aboral axis is regulated by zygotic Wnt/ β -catenin gradient, just as it is in bilaterians. This paper provides critical information that will significantly modify views of how early development evolved in the animal kingdom and suggests that endomesoderm specification by Wnt/ β -catenin signaling may be an evolutionary innovation limited to bilaterians. This manuscript will be highly interesting to those investigating how cell signals influence evolution of animal body plans.

Comments for transmission to the authors

I appreciated having the opportunity to review this paper. I found it exciting and think that it makes an important novel contribution to evodevo studies. The authors have been thorough in their investigations and show an impressive knowledge of the literature related to this study. As I mentioned above, this paper provides two especially noteworthy contributions. First, the authors have demonstrated that endomesoderm forms at the pole opposite of the cells in which maternal nuclear β -catenin signaling is occurring, therefore the maternal Wnt signal is not responsible for specifying endomesoderm in this group. Second, they demonstrate that a gradient of zygotic β -catenin signaling is influencing cell fates in the specification of the aboral-oral axis in a manner similar to bilaterians. Both findings involved careful analysis and this group seems to have done this well using sound methodology and meeting the expected standards in my field. In most cases, I thought that the authors did a nice job supporting their observations, that their review of the literature was thorough, and that the paper was clearly written. I've made a few suggested improvements below. I sometimes found this study confusing because at least 12 markers and numerous perturbation methods were used. Following this was sometimes difficult so I've included suggestions to help readers with this challenge.

- Line 25, page 2...you said that JNK signaling appears to act agonistically with beta-catenin signaling. Is that what you meant or did you mean that the JNK inhibitor acts agonistically with beta-catenin signaling, since the JNK inhibitor counteracts the effects of activated Wnt signaling?

Since JNK inhibition has an effect opposite to Wnt upregulation by azakenpaullone, it suggests that JNK and Wnt signalling act “in the same direction”, so, agonistically.

- When you first mention use of azakenpaullone (line 28, page 2), alsterpaullone (line 18,

page 4) and iCRT14 (line 28, page 9) it may help to also refer the reader to Figure 1b. Or at least do this when you mention the first inhibitor, azakenpaullone.

Done.

- Lines 11-20, page 7...Elaborate on the specific evidence that contradicted those findings by mentioning the specific markers that were expressed for each group discussed (e.g. Snail, ERP, etc.). In Line 13, why not say ectodermal markers instead of “zygotic markers of the aboral/anterior/low β -catenin-signalling end of the O-A axis?” In the sea urchin the markers expressed would be ectodermal.

Corrected as suggested.

- Line 29, page 9...Elaborate a little on why the presence of a shortened pharynx and expanded gut in cydippids suggests an “antiendomesodermal” role for beta-catenin to make it easier for the reader to understand why you came to this conclusion.

Done.

- When discussing the distribution of mesodermal gene expression, you mentioned ERG and snail. It might be good to also cite two of the Snail papers for readers interested in a closer look at snail distribution. I'd cite Fritzenwanker, Saina, and Technau, 2004. Analysis of forkhead and snail expression reveals epithelial–mesenchymal transitions during embryonic and larval development of *Nematostella vectensis*. as well as Magie, Daly, and Martindale, 2007. Gastrulation in the cnidarian *Nematostella vectensis* occurs via invagination not ingression. Both provided additional helpful information about expression of this mesodermal marker.

Done.

- Figure 2... In 2a, remind the reader that they're looking at the animal (oral) and vegetal (aboral) poles (under “V” you could write (aboral) and under “An” you could write (oral)). Because it's necessary to switch back and forth from saying aboral/oral, animal/vegetal, and posterior/anterior, it can be confusing. This modification may help the reader a little.

Thank you for the suggestion. There was no place on the figure, but we specified it in the legend.

- Figure 2...What stages are shown in Figure 2k-o? In 2a-j, you clearly indicate the age of the embryo and what time the photo was taken. Please also do this for the figures below those. Also, please explain notable features in f-j and k-o (I believe k-o shows cells that are forming mesoderm. How did you decide which cells were mesodermal in k-o?).

The times shown of images Fig. 2a-f are average developmental ages we determined for specific rounds of cleavage in Fritzenwanker et al., 2007 for embryos developing at 18°C and not embedded in agarose. This is mentioned in the legend and the reference to the Fritzenwanker et al., 2007 is provided. The speed at which embedded embryos were developing while being scanned under the microscope was much faster because the temperature of the room with the confocal microscope was ~23° at the start of the imaging and, likely, slightly higher towards the end of the imaging, as the cooling system of the

microscope heated the room up. It must be noted, however, that higher temperature is not a concern: in nature, *Nematostella* polyps live in shallow brackish pools, which get extremely warm. The standard spawning induction procedure is by placing tanks with the polyps kept at 18°C into a 25°C incubator (Fritzenwanker et al., 2002), and we also routinely let the embryos develop in the 26°C incubator if we need to reach certain developmental stages or let the polyps develop to sexual maturity quicker. At 26°C, the embryos develop normally, but faster.

Movies S1 and S2 contain 91 frames taken every 5 minutes, which accounts for 28823.04 seconds (slightly over 8 hours) of development and encompass development from 32-cell stage until the onset of gastrulation. This shows that the embryos started gastrulating several hours earlier than they would do at 18°C. Movies S3 and S4 contain 102 frames covering 32422.34 seconds (9 hours) of development. In our opinion, indicating these times on the images would be misleading, so we chose to show average times at which embryos reach corresponding developmental stages in regular culture conditions, as we defined in Fritzenwanker et al., 2007. Indicating average developmental stages/times for Fig. 2g-o is, unfortunately, not possible because the embryos lack clear morphological landmarks at the this stage, and the number of cells can only be estimated based on changes in cell size between the 1024-cell stage on Fig. 2f and approximately 5000 cells in the early gastrula on Fig. 2o (the number of cells in a mid-gastrula of *Nematostella* is approx. 7000, as shown by Kirillova et al., 2018). In the revised manuscript, we added a Supplementary Figure with images from Fig. 2a-o with time stamps extracted from the confocal metadata. Time 00:00:00 corresponds to the first frame of the Movie S1 (32-cell stage)

Prospective mesodermal cells on k-o were identified as the ones forming apical constrictions. Additionally, to independently confirm that the β -catenin negative cells on images c-f are indeed the ones making the mesoderm later, we performed in situ hybridization using HCR probes against the early mesodermal marker *ERG* combined with an anti-GFP staining. These results are presented on the new Supplementary Fig. 5.

- Extended Data Figure 1...Remind the reader that JNK-IN-8 is a JNK inhibitor and explain how you to expect it to affect Wnt signaling in the caption. Also, for all of the figures include the age or stage of the embryos shown and mention the temperature that embryos were cultured at in the Methods.

Done.

- In the figure legends, please clearly define what the markers are as you did in Extended Data Figure 3. There are so many markers used in your paper and it can be difficult to remember which tissues each one marks. I spent a lot of time looking this information up for the different figures. For example define the markers in Extended Data Figure 1...In the paper, you mentioned that Bra is an oral marker, Wnt2 is a midbody marker, and Six3/6 is an aboral marker. Also tell the readers what snailA and foxA indicate.

Done

- In Extended Data, Fig. 4c-f, do all 4 photos show sea urchin embryos? What species is it? Readers unfamiliar with these embryos may get confused and think they're comparing *Nematostella* to sea urchin.

Corrected, thank you!

- Extended Data, Figure 5...State that the embryo in the figure is *Nematostella* (I assume it is, but this wasn't mentioned).

Corrected

Also, in the Wikramanayake 2003 paper, the authors blocked Wnt signaling by overexpressing cadherin. It would have been nice if you had used the same perturbation in this paper or explained why you chose to use RNAi LRP5/6. I expect both perturbations will produce the same phenotype but would have found it more convincing if the same perturbation had been used. Include citations supporting this as a method of blocking Wnt signaling and address whether this perturbation is expected to produce the same phenotype that overexpressing cadherin would have.

Wikramanayake et al used cytoplasmic domain of the cadherin from the sea urchin *Lytechinus variegatus* to titrate away *Nematostella* β -catenin. We do not have access to this reagent, however, we think that using a *Nematostella* β -catenin-specific reagent is a better option. We used one of the two β -catenin translation blocking morpholinos previously characterized by Leclere et al., 2016 and used by us in earlier work (e.g. in Lebedeva et al., 2021). This morpholino produces the same morphological phenotype as shown in Wikramanayake et al., 2003, and the paper of Leclere et al., 2016 is cited. The same phenotype is also reproduced if we inject mRNA encoding the intracellular domain of the *Nematostella* Cadherin 3 (see Figure below; *ERG* expression in the endoderm and aboral ectoderm observed in the control embryos starts much later than its mesodermal expression).

Any of these perturbations of β -catenin signaling result in a gastrulation failure and ubiquitous expression of the mesodermal markers in the embryo. However, by directly preventing the function of β -catenin, Wikramanayake et al. not only suppressed Wnt signaling but also the activity of maternally deposited β -catenin (see new Supplementary Fig. 8b), which, as we have shown previously (Niedermoser et al., 2022) and here (Supplementary Fig. 8a) is not affected by Wnt. Wnt-mediated β -catenin signaling requires LRP5/6. In the Niedermoser et al, 2022, we knocked LRP5/6 down with two different shRNAs and with a morpholino, and in all cases we showed that endomesoderm specification was not prevented, however, the oral-aboral patterning failed completely: endodermal and midbody markers were abolished, and all non-mesodermal cells were expressing anterior/aboral ectoderm markers. This allowed us to conclude that Wnt/LRP5/6/Frizzled-mediated β -catenin signaling was not required for the endomesoderm specification but it was essential for axial patterning (Niedermoser et al., 2022). Here we show that, in line with our earlier findings, LRP5/6 knockdown does not prevent the accumulation of nuclear maternal β -catenin on the vegetal/aboral/anterior pole of the embryo as well its absence in the future mesodermal domain (Supplementary Figure 8).

Reviewer #2 (Remarks to the Author):

The manuscript ‘ β -catenin-driven endomesoderm specification is a Bilateria-specific novelty’ by Lebedeva et al reinvestigates and reassesses the role of beta-catenin in early endoderm specification of the sea anemone. Intriguingly, they observe an early aboral nuclear beta-catenin localization opposite to the oral ‘endomesoderm’. This finding may suggest that β -catenin represses endomesoderm specification in the sea anemone, rather than activating it as seen in bilaterians challenging the established paradigm of beta-catenin function in the cnidarian/bilaterian ancestor.

The authors used CRISPR-Cas9-mediated genome editing to generate a knock-in line tagging the endogenous beta-catenin gene with a Nterminal superfolder GFP in the sea anemone. They used these transgenic animals to first show that there is indeed a weak oral to aboral bimodal beta-catenin gradient at the gastrula stage that correlates with axin expression, responds to different concentration of the GSK3 inhibitor (ALP), and leads to expression changes of target genes. These findings are consistent with previous work and the assumption that an oral to aboral beta-catenin signalling gradient controls domains along the O-A axis.

The authors then focused on the earlier event during the time of endomesoderm specification during the mid-blastula stage. They showed by in situ in mid-blastula stages a domain of cells that coexpress mesodermal (erg) and endodermal (bra, wnt1, wnt3, wntA) markers. Surprisingly, in video recordings of immobilized embryos at earlier stages nuclear sfGFP-beta-catenin was visible in a large domain opposite to the later oral side of gastrulation and mesoderm formation. Early incubation with the GSK3beta inhibitor (ALP) led to strong nuclear localization of sfGFP-beta-catenin in all mid blastula cells.

Furthermore, crossing reciprocally homozygous sfGFP-beta-catenin female or male to a wildtype male or female, respectively, the authors could demonstrate that the early aboral nuclear expression is indeed maternal, and the later O to A gradient zygotic.

Based on these patterns the authors conclude that early beta-catenin signaling happens aborally/vegetally apparently opposite to the area of endomesoderm specification on the oral

side. This pattern is obviously not consistent with the previous hypothesis that endomesoderm specification is specified by an early beta-catenin patterning as observed in many bilaterian embryos.

The authors then argue that previous experimental observations and the observed early aboral pattern of nuclear beta-catenin support rather a mode of early beta-catenin repression of endomesoderm and later mesoderm specification in the sea anemone.

Although the authors show beautifully the nuclear patterning of beta-catenin by using a state of the art knock in tagging strategy, and show convincingly an early aboral pattern of nuclear beta-catenin localization contradicting the currently assumed role of beta-catenin in endomesoderm specification in cnidarians, I have several concerns in regard to some of the interpretation, and the scope and strength of the experimental support. If one wants to make a paradigm changing statement, the authors need to show stronger evidence for endomesoderm repression by early beta-catenin.

Main concern 1:

Indeed I do agree that the beta-catenin morphant phenotype described in Figure 2 t1 (circumferential mesoderm marker expression) might consistent with this new repression model but I don't understand how it can be consistent with the early beta-catenin activation phenotype in Figure 2 t2 (ubiquitous endoderm marker expression)? Would one not expect no mesoderm, and no endoderm upon early beta-catenin activation? If the scenario presented in Fig 3b for the sea anemone is right, should not early beta-catenin activation lead to a complete loss of endomesoderm, and an expansion of the anterior ectoderm/aboral marker genes similar to the beta-catenin suppression phenotype in sea urchin (Figure 2 u1)?

Similarly in the discussion, quote: "Finally, the fact that β -catenin inhibits endomesoderm and, subsequently, mesoderm specification rather than activates it explains why upon treatment with a GSK3 β inhibitor prior to 6 hpf, the whole embryo acquires endodermal rather than mesodermal or mixed endodermal and mesodermal fate." I do not understand this conclusion. Should not - based on regulatory logic of a repression model - as a consequence of beta-catenin activation by early GSK inhibition, the embryo be mostly expanding the aboral area, and not expanding the endoderm? How can a ubiquitous repression of endomesoderm specification lead to endoderm? See also extended figure 4: a4 makes sense, but not the a3 result?

Thanks for this comment, we see the logic of the Reviewer, but the situation is more complex. Similar to the situation in sea urchin, ectodermal cells in *Nematostella* can be re-programmed to acquire endodermal fate by stabilization of β -catenin, therefore, expansion of the endodermal markers is exactly what is expected. Since *Nematostella* endomesoderm and, subsequently, mesoderm is repressed by β -catenin, β -catenin morphants express mesodermal markers ubiquitously.

The difference between *Nematostella* and sea urchin is in how the endomesodermal and, subsequently, mesodermal domain is specified. In sea urchin, endomesoderm is specified cell-autonomously by maternal β -catenin. Shortly after specification, endomesoderm segregates into endoderm, which remains β -catenin-positive, surrounding the mesoderm, which becomes β -catenin-negative. Due to the cell-autonomous nature of mesoderm specification, stabilization of β -catenin cannot convert mesoderm to endoderm. The situation in *Nematostella* is different. Endomesoderm specification can be prevented by β -catenin

stabilization before 6 hours post-fertilization (as on Fig. 2 t2 and Supplementary Fig. 6a3), but not after 6 hpf, when endomesoderm is already defined and cannot be reprogrammed (Supplementary Fig. 6a4, Niedermoser et al., 2022). We show here (Fig. 2, Supplementary Fig. 5) that in contrast to sea urchin, *Nematostella* endomesodermal cells do not contain maternal nuclear β -catenin, and analysis of β -catenin morphants (Supplementary Fig. 7; Leclere et al., 2016; see also new preprint of Haillot et al., 2024 for more mesodermal markers <https://doi.org/10.1101/2024.10.29.620801>) clearly shows that mesodermal markers are repressed by β -catenin signaling.

Reviewers' suggestion that upon β -catenin the embryo should express anterior ectoderm/aboral markers is logical but not entirely correct, because the effect of β -catenin on the expression of same genes is different at different developmental time points. qPCR data on embryos with up- or downregulated β -catenin signaling presented in this paper (see new Supplementary Fig. 7) suggest that maternal nuclear β -catenin signaling does not seem to significantly affect endodermal gene expression in 8-9 hpf embryos (see values on the y-axis on the Supplementary Fig. 7 and compare to the expression levels of the mesodermal genes). Moreover, Haillot et al., 2024 showed that ectodermal markers *APC* and *koza-like* start to be expressed ubiquitously upon GSK3 inhibition and abolished in β -catenin morphants, which is exactly the opposite expression behavior compared to mesodermal genes. Thus, at 8-9 hpf, the embryos behave just the way the Reviewer suggested.

The situation changes as maternal β -catenin disappears and endodermal marker expression is turned on around 10-12 hpf first in the endomesoderm and then in the surrounding ectodermal cells, which become the definitive endoderm giving rise to the blastopore lip and pharynx. This onset of endodermal gene expression appears to depend on Delta/Notch signaling (Haillot et al., 2024), but once endodermal gene expression starts, endodermal cells and, importantly, also **the ectodermal cells** start to respond to Wnt/ β -catenin signal. Since, as we have shown earlier, orally expressed β -catenin-dependent transcription factors are capable of repressing more aborally expressed β -catenin-dependent genes (Lebedeva et al., 2021), ectodermally expressed genes become repressed in later stage embryos upon activation of β -catenin signaling by GSK3 inhibitor treatment, while endodermal markers become ubiquitously expressed.

Taken together, β -catenin upregulation can activate endodermal gene expression in the ectodermal cells of the embryos older than 12 hpf but not in the younger embryos. In contrast, β -catenin downregulation results in the ubiquitous de-repression of the fate, which is normally suppressed by β -catenin signaling. Both, mesodermal markers and late anterior/aboral ectoderm markers (*Six3/6*, *FoxQ2a*) are suppressed by β -catenin, however, since something in the mesoderm (most likely MAPK signaling – see Haillot et al., 2024) suppresses anterior ectoderm fate, the whole β -catenin morphant embryo acquires mesodermal identity. We edited Supplementary Text to make this clearer.

Main concern 2:

In this context I also find the description in the abstract too strong “we show that, in contrast to bilaterians, *Nematostella* endomesoderm specification is repressed by β -catenin”. While I agree with the reinterpretation of the beta-catenin morphant phenotype, although there is the possibility that beta-catenin morpholinos did not interfere with maternal beta-catenin protein, the authors need to show much stronger experimental evidence for repression with additional markers in early mid blastula stages. Are there any early aboral markers that could be used,

and is there a way to elicit and/or inhibit beta-catenin activation in distinct early blastomeres, and look at the response of marker genes? There is no direct evidence for repression.

We thank the Reviewer for this comment, and we will try to address it point-by-point.

1. β -catenin morpholino cannot interfere with maternal β -catenin protein, which, we know, exists in the embryo since unfertilized eggs laid by sfGFP- β -catenin females glow green. However, other methods of β -catenin inhibition, which directly target β -catenin protein rather than β -catenin mRNA, such as titrating away β -catenin by overexpression of the intracellular domain of *Nematostella* Cadherin 3 (see Figure in the response to Reviewer 1), phenocopy the β -catenin morpholino phenotype.
2. In the first version of the paper, we were unable to show an array of mesodermal marker gene de-repression in β -catenin morphants because we knew that these analyses were being done by a postdoc in the group next door, and this would have scooped a good colleague. Now, the analysis of Haillot et al., is available on bioRxiv, and their findings are in line with ours (despite minor semantic differences in the naming of the germ layer identities). Here is a short summary of the Haillot et al. and our findings:

Detailed scRNA-Seq analyses followed by in situ hybridization characterization of a number of individual candidate genes performed in Haillot et al., 2024 (<https://doi.org/10.1101/2024.10.29.620801>) clearly show that mesodermal markers are the first zygotic germ layer-specific markers to be activated in the embryo in the endomesodermal domain. Several hours later, the same cells, which express mesodermal markers, start to express endodermal markers (*Brachyury*, *Wnt1*, *Wnt3*, *WntA* - see Supplementary Fig. 3), which allows us to call this transient state “endomesoderm”. Subsequently endodermal marker expression moves out into the definitive endodermal domain surrounding the cells expressing mesodermal markers (see this paper as well as Kraus et al., 2016). All mesodermal markers analyzed by in situ hybridization (*fgf1*, *hand2*, *tbx19*, *snailA*, *zicA*, *Nkx2.2D*, *Nkx2.2B*, *mitf-like*, *smad1/5* - see Haillot et al., 2024, and also *ERG* – see figure in the response to the first Reviewer), become ubiquitously expressed in the β -catenin morphants and abolished in embryos with β -catenin signaling pharmacologically activated by GSK3 inhibition. In contrast, in 8 hpf embryos – i.e. shortly after the onset of the mesodermal gene expression, the earliest known ectodermal markers *koza-like* and *APC* fail to clear out of the endomesodermal domain if β -catenin is stabilized by a GSK3 inhibitor azakenpaullone, and, conversely, *APC* and *koza-like* expression is abolished in β -catenin morphants. This suggests that strong maternal β -catenin we observe on the vegetal side of the embryo is responsible for the activation of the early ectodermal markers like *APC* and *koza-like* and for repression of the mesodermal markers outside the endomesodermal domain during early developmental stages (until ~12 hpf). These observations are fully in line with our statement that endomesoderm specification is repressed by β -catenin. In the revised version of our paper, we refer to the work of Haillot et al., 2024

To follow on the Reviewers’ request to analyse the expression of endodermal markers, which are just starting to be expressed, in early mid-blastula stages, we turned to using qPCR and added a new Supplementary Fig. 7 with qPCR analysis of expression of 3 mesodermal markers, 6 endodermal markers, and 3 zygotic markers of

aboral/anterior ectoderm in β -catenin morphants and in GSK3 inhibitor-treated embryos at 9 and 12 hpf. qPCR data fully confirms the repressive effect of β -catenin on mesodermal marker gene expression shown by Haillet et al., 2024 in their in situ hybridization analyses. Strikingly, endodermal genes, all of which are known to be β -catenin-dependent several hours later in development (see Lebedeva et al., 2021) appear not to be sensitive to β -catenin signaling at this early stage. Thus, their expression must be activated by some other signal, possibly by Delta/Notch, as the findings of Haillet et al., 2024 suggest – (Delta is a mesodermal marker, and Notch is initially ubiquitous and then becomes suppressed in the mesoderm).

Main concern 3:

To support the authors hypothesis that endomesoderm specification by beta-catenin is a bilaterian novelty the authors show one ctenophore experiment. The authors argue that an early pharmacological inhibition of beta-catenin in a ctenophore leads to more gut/endodermal tissue consistent with their repression hypothesis. Indeed, the internal tissues including the Lysotracker positive endoderm seem to be expanded after treatment. However, this is by all means fairly weak supporting evidence, and an isolated experiment, that need to be corroborated by more evidence e.g. additional markers, and experiments. I don't think it should be used at this stage as supporting evidence. This phenotype needs to be characterized better.

We agree that we do not have proof that in ctenophores beta-catenin plays a role in suppressing endomesoderm or mesoderm formation, however, we also do not present it as such. We bring it up in the Discussion in a paragraph starting with a sentence “In order to verify that endomesoderm specification by β -catenin is indeed a bilaterian novelty it will be important to address the mechanisms of endomesoderm specification in the representatives of earlier branching animal clades” and end the paper with the question “iii) does β -catenin signalling really prevent endomesoderm formation in the earlier branching Ctenophora, which also gastrulate from the animal pole?”.

First author of this paper, Tatiana Lebedeva, has just started a postdoc devoted to exactly this question. However, we think that this discussion point is important, the pilot experiment provided us with some data, and we discuss it together with the data of Salinas-Saavedra et al., which contradicts our finding. We do not see any reason to remove either of these.

Minor concerns:

1. Figure 2 a to o: How do the authors discern the animal and vegetal pole here? Based on the location of polar bodies, or retroactively based on the later animal location of the mesoderm? Is the pink area in later stages (prospective mesoderm) discerned based on morphology and shape of the cells? What is the localization of the polar bodies (also in the movies)? The authors should show early nuclear beta-catenin localization together with early endomesodermal and/or anterior ectoderm markers.

In all Cnidaria, polar bodies are located next to germinal vesicle at the animal/oral/posterior pole. Previous work by us and others (Fritzenwanker et al., 2007; Lee et al., 2007) showed that gastrulation in *Nematostella* takes place at the animal pole. However, in *Nematostella*,

polar bodies are usually shed off as the egg is pressed out of the gonad, so they cannot be seen in the movies. Fig. 2 shows individual frames of the Supplementary movie 1. We did not know where the oral and aboral poles were initially, but the embryos are embedded in agarose and do not turn. We followed the developing embryos frame-by-frame, and we know which cells contribute to the invaginating mesodermal plate at the onset of gastrulation. The pink area was easily discerned morphologically as cells undergoing apical constriction and subsequently invaginating. We added a note in the Fig. 2 legend indicating that mesodermal cells were highlighted based on morphology.

In the revised version of the paper, we add Supplementary Fig. 5, in which we show that strong sfGFP- β -catenin signal is observed in the interphase nuclei of the cells, which do not express the mesodermal marker *ERG* at 9 hpf – fully in line with the data shown in the Supplementary Movies 1 and 2 and Fig. 2a-o. In contrast, at 12 hpf, strong early sfGFP- β -catenin signal progressively disappears, and weaker nuclear sfGFP- β -catenin signal start in the cells on either side of the *ERG* expression boundary

2. Figure 2 p: why is there circumferential localization of beta-catenin? It should be absent on one side. I assume the authors will argue that it is the section but I would prefer a section that shows both the activation only in one part but not the other as less misleading.

It is indeed the section. The point of Fig. 2p is to show that strong early sfGFP- β -catenin signal is maternal, and the figure shows it, since at 9 hpf nuclear staining is only observed in the embryos developing from eggs laid by transgenic mothers. For embryos with a clear gap in the sfGFP- β -catenin staining in the endomesodermal domain please see Supplementary Movies 1 and 2 as well as the Supplementary Figures 5 and 8 showing maximum intensity projections of z-stacks of 9 hpf and 8 hpf embryos, respectively.

3. Figure 2c: The weak nuclear beta-catenin is difficult to discern from this section e.g. one can see some nuclei on left side of the gastrula but not on the right. Why didn't the authors normalize the nuclear GFP signal using nuclear DNA staining?

Fig. 2c shows a very early blastula live-imaged at the 128-cell stage. It is a frame from the Supplementary movie 1, and exposure is the same in all frames. Weak sfGFP- β -catenin signal is biologically relevant, it only becomes strong in the next cell division.

3. Extended Figure 3: d to g

The *ERG* positive territory in the 12hpf sea anemone embryo has a very different appearance and shape in each stained embryo. Would one not expect a more regular shaped appearance of this domain based on similar endomesoderm-like territories in other embryos? It is irritating that the orientation for each double in situ is all over the place. This irregular patch of 'endomesoderm' in wildtype sea anemone seems kind of strange. However, I am also not that familiar with *Nematostella* embryos.

Nematostella endomesoderm or pre-endodermal plate, as it has been termed in older papers, always has an irregular shape. It can be triangular, square, polygonal, star-shaped etc. See for example Kraus et al., 2006. We added a note to the figure legend.

4. Extended Figure 3: d to g, c to h

It is somewhat unclear how the early ‘endomesoderm’ patch relates to later mesoderm and endoderm in *Nematostella*. Does the endomesoderm area differentiate into mesoderm in gastrula stages or into both the endodermal ‘ring’ and the mesoderm? Or is the endoderm marker expression somewhat outside the earlier ‘endomesoderm’ patch?

Our current understanding is that early endomesoderm is differentiating into mesoderm, and endodermal markers move out of the endomesoderm into the surrounding cells. Haillot et al., 2024 suggest that definitive endoderm is induced by the mesodermal signal in a single row of ectodermal cells abutting the mesoderm. This is especially easily visible in the *Bra/ERG* double in situ, but also in case of *Wnt1* and *WntA*, but not in *Wnt3*, which exits the mesodermal territory later. Green *Bra* staining extends outside the magenta *ERG* domain into the definitive endoderm. We changed the sketch on Supplementary Fig. 3c to clarify this and added an explanation to the legend.

5. Extended data Fig. 2: It looks like the size of embryos was really affected by the treatment (compare h to i).

Nematostella embryos differ very significantly in size even if they come from the same egg package laid by the same female. The reason for that is the huge variability of egg size (*Nematostella* egg size varies easily by 25-30%). Therefore, we are certain that the size difference is completely normal and is not due to treatment (compare h to i to j).

6. Lane 24: ‘Midbody’ domain: misleading term consider other name

We used this name because we defined these terms in the Lebedeva et al. 2021 paper and want to keep it consistent. We added the reference to the paper.

7. Page 2 [15-20] Rephrase this complex sentence.

We removed the end of the sentence.

8. Page 4 [15] Point out the small peak at the border between the midbody and the aboral domain (approximately at relative position 0.60) in Fig 1e.

Done.

Reviewer #3 (Remarks to the Author):

In this article, Lebedeva et al. analyzed the role of b-catenin signaling in endomesoderm specification and axial patterning in the Cnidaria *Nematostella vectensis*. They characterized b-catenin localization by tagging the endogenous protein with GFP and also tested the consequences of up- or downregulation of b-catenin signaling. Their data confirm the role of b-catenin in axial patterning that was previously suggested. More interestingly, they observed that b-catenin is excluded from the early endomesoderm territory, contrary to what was previously assumed, suggesting that b-catenin driven endomesoderm specification is a novelty of Bilateria.

The article is overall well written and the data are convincing. However, a few points could be improved before publication:

- the authors claim that *Nematostella* endomesoderm is repressed by b-catenin. However, I think the authors could provide better evidence to support this claim. The fact that b-catenin is excluded from the early endomesoderm territory doesn't necessarily mean that b-catenin is repressing it. They show the effect of b-catenin depletion or upregulation on the expression of markers at a late stage when mesoderm and endoderm are already separated. b-catenin depletion (b-catenin MO) expands mesoderm but it also represses endoderm. b-catenin upregulation (AZK) represses mesoderm but it also expands endoderm. To me these observations at a late stage make the interpretation in term of "endomesoderm specification" a bit difficult. The authors nicely define endomesoderm at 12 hpf blastula stage (an earlier stage) as a territory that coexpresses endodermal and mesodermal markers (extended data Fig. 3d-g). I think they should analyze the effect of b-catenin depletion or upregulation on the co-expression of these markers at this stage (12 hpf blastula).

Thank you for this comment, we agree. Morphologically, *Nematostella* is a diploblastic animal, so the terms "ectoderm", "endoderm" and "mesoderm" refer to molecular identities of these embryonic tissues. In the revised version of the paper, we show qPCR data on 9 hpf and 12 hpf blastulae upon β -catenin depletion and upregulation showing that β -catenin signaling indeed suppresses mesodermal marker expression (Supplementary Fig. 7). Moreover, endodermal markers are not responsive to alterations in β -catenin signaling intensity. These data are independently confirmed by a thorough in situ hybridization based study by Haillot et al., 2024, which has recently become available on bioRxiv at <https://doi.org/10.1101/2024.10.29.620801>.

Please also see our response to the major concern 2 of the Reviewer 2.

- it would be useful to have a definition of the terms "animal-vegetal axis" and "oral-aboral axis" at the beginning of the manuscript.

We tried to add an explanation without disrupting the flow of the text.

REVIEWERS' COMMENTS

Reviewer #1 (Remarks to the Author):

I approve of the changes that were made and appreciated the authors' explanations concerning why they chose not to modify in a few cases (I support their decisions). I look forward to seeing this manuscript published in the journal.

Thank you for your time and suggestions to the first version of the paper.

Reviewer #2 (Remarks to the Author):

I thank the authors for carefully and patiently addressing my previous concerns, and putting such a clear, rigorous, and exciting revision together. No objections and concerns.

Thank you for your time and suggestions to the first version of the paper.

Minor error:

Page 6 line 17: much greater impact?

However, our second observation that *Nematostella* endomesoderm forms in the β -catenin negative domain has a much greater import for the understanding of the early evolution of the body axes and germ layers.

Corrected, thank you!

Reviewer #3 (Remarks to the Author):

The authors have addressed all the points that I raised in a satisfactory manner.

Thank you for your time and suggestions to the first version of the paper.